# Approaches Used to Construct Antibiograms for Dogs in a Veterinary Teaching Hospital in the United States

**DOI:** 10.3390/antibiotics12061034

**Published:** 2023-06-09

**Authors:** John E. Ekakoro, Lynn Guptill, Kenitra Hendrix, Melinda Anderson, Audrey Ruple

**Affiliations:** 1Department of Public and Ecosystem Health, College of Veterinary Medicine, Cornell University, Ithaca, NY 14853, USA; jee94@cornell.edu; 2Department of Veterinary Clinical Sciences, College of Veterinary Medicine, Purdue University, West Lafayette, IN 47907, USA; guptillc@purdue.edu; 3Indiana Animal Disease Diagnostic Laboratory, Department of Comparative Pathobiology, College of Veterinary Medicine, Purdue University, West Lafayette, IN 47907, USA; gkh@purdue.edu; 4Department of Basic Medical Sciences, College of Veterinary Medicine, Purdue University, West Lafayette, IN 47907, USA; andersmj@purdue.edu; 5Department of Population Health Sciences, Virginia-Maryland College of Veterinary Medicine, Virginia Tech, Blacksburg, VA 24061, USA

**Keywords:** antimicrobial resistance, canine patients, empiric antimicrobial therapy, Gram-negative bacteria, Gram-positive bacteria, veterinary antibiograms

## Abstract

Non-judicious antimicrobial use (AMU) is a major driver of antimicrobial resistance (AMR). In human hospitals, cumulative antibiograms are often used by clinicians to evaluate local susceptibility rates and to select the most appropriate empiric therapy with the aim of minimizing inappropriate AMU. However, the use of cumulative antibiograms to guide empiric antimicrobial therapy in veterinary hospitals in the United States is limited, and there are no specific guidelines or standardized methods available for the construction of antibiograms in veterinary clinical settings. The objective of this methods article is to describe the approaches that were used to construct antibiograms from clinical samples collected from dogs seen at a veterinary teaching hospital. Laboratory data for 563 dogs for the period from 1 January 2015 to 31 December 2015 was utilized. We used the Clinical and Laboratory Standards Institute (CLSI) guidelines for use in the construction of the antibiograms in human healthcare settings as the basis for the veterinary antibiograms. One general antibiogram and antibiograms stratified by hospital section, the anatomic region of sample collection/by sample type, were created and the challenges encountered in preparing these antibiograms were highlighted. The approaches described could be useful in guiding veterinary antibiogram development for empiric therapy.

## 1. Introduction

Antimicrobial resistance (AMR) is a serious healthcare problem that is increasing in incidence worldwide [1], and it is a threat to global health [2]. Any use of antimicrobials, whether judicious or non-judicious, creates selection pressure that can lead to AMR emergence [3]. However, non-judicious use of antimicrobials in human and veterinary medicine is a modifiable driver of AMR, leading to the emergence of bacterial strains resistant to many antimicrobials [4,5,6]. Like in human medicine [7], initial antimicrobial therapy in veterinary medicine is often empiric and is guided by clinical signs of disease. Under ideal clinical practice situations, empirical use of antimicrobials is discouraged and antimicrobial prescription based on a bacterial culture and antimicrobial susceptibility testing is preferred [8]. However, high costs and long turnaround times associated with the existing bacterial culture and antimicrobial susceptibility testing methods influence the decision to use them [9,10].

In human hospitals, summaries of the patterns of antimicrobial susceptibilities (antibiograms), constructed from laboratory samples that have been submitted institutionally are often used by clinicians. The antibiograms report the percentage of isolates susceptible to a set of antimicrobials, may be stratified (by patient population, medical service or specimen type), and are shared with clinicians, hospital pharmacists and infection-control personnel in the form of web postings and pocket guides [11]. These antibiograms can complement rapid diagnostic techniques in clinical microbiology [12], allow clinicians to evaluate the local susceptibility rates to specific antimicrobials and aid them in selection of the most appropriate empiric therapy [13,14,15].

The integration of antibiogram use in veterinary practice to guide empiric antimicrobial therapy is necessary for antimicrobial stewardship [16]. However, in the United States, the use of cumulative antibiograms to guide empiric antimicrobial therapy in veterinary medicine is still limited to just a few tertiary veterinary hospitals. Even the two tertiary veterinary hospitals that in the past publicly provided cumulative antibiograms to their clinicians via their hospital websites do not have up-to-date antibiograms posted on their websites. The limited use of cumulative antibiograms in veterinary medicine may be due in part to the fact that there are no specific guidelines or standardized methods available for the construction of veterinary antibiograms. Here, we examined the practicality of constructing cumulative antibiograms for a veterinary tertiary care hospital. This paper describes and documents the approaches used, and the challenges that were encountered in constructing cumulative antibiograms for a veterinary teaching hospital in Indiana, the United States.

## 2. Results

### 2.1. The Isolates

Of the 563 canine antimicrobial susceptibility profiles in the analyzed dataset, 318 (56.5%) isolates were Gram-positive and these included *Staphylococcus* spp., *Enterococcus* spp. *Streptococcus* spp., *Corynebacterium* spp., *Bacillus* spp., *Aerococcus* spp., and Actinomyces spp. Two hundred forty-five isolates (43.5%) were Gram-negative and these included *Escherichia coli*, *Pseudomonas* spp., *Proteus mirabilis*, *Pasteurella* spp., *Enterobacter* spp., *Klebsiella* spp., and other Gram-negative organisms.

### 2.2. General Antibiograms

One general antibiogram with a section for Gram-negative organisms (Table 1) and another for Gram-positive organisms were prepared (Table 2). The Gram-negative isolates were 100 *Escherichia coli*, 27 *Pseudomonas* spp., and 26 *Proteus mirabilis* (Table 1). For the Gram-positive isolates, there were 170 *Staphylococcus* spp. isolates, 64 *Streptococcus* spp., 30 *Corynebacterium* spp., and 65 *Enterococcus* spp. isolates (Table 2).

### 2.3. Antibiograms Stratified by Hospital Section (Medical Service), Sample Type/Anatomic Region of Sample Collection

There were enough isolates collected from different hospital sections, the ears, skin, and urine to create individual antibiograms for the different hospital sections and anatomic regions/sample types. Antibiograms with sections for patients admitted to each hospital department were prepared for Gram-positive and Gram-negative organisms (Table 3, Table 4, and Table 5, respectively). Twenty-one *Staphylococcus* spp. isolates from the ear and 43 Staphylococcal isolates from the skin were used to create antibiograms for these anatomic regions (Table 6). One antibiogram with two parts, one for Gram-negative and the other for Gram-positive bacteria, was prepared for the isolates from urine (Table 7)

## 3. Discussion

This article adds to the literature a description of practical approaches used in making antibiograms for a veterinary teaching hospital. This work shows that creating cumulative antibiograms for dogs seen at a veterinary teaching hospital stratified by area of hospital admission and sample type is feasible and practical. Similar to ours, another study conducted in the US demonstrated the feasibility of adapting the existing guidelines for developing antibiograms in human medicine to the veterinary companion animal private practice setting [17]. The results of these antibiograms show the tremendous variability in the antimicrobial susceptibilities of the isolates to specific antimicrobials at a veterinary teaching hospital in Indiana, e.g., in the general antibiograms, 90% of the staphylococcal isolates were susceptible to amoxicillin–clavulanic acid, only 70% were susceptible to enrofloxacin, 73% of *E. coli* were susceptible to amoxicillin–clavulanic acid and 85% were susceptible to enrofloxacin. These susceptibility differences suggest that antibiograms could be very useful in guiding empiric antimicrobial therapy and could guide the framing of antimicrobial stewardship policies at this hospital.

The *E. coli* isolated from samples obtained from the Medicine service showed susceptibility profiles similar to those observed in the general Gram-negative antibiogram. Of the 43 *Staphylococcus* spp. isolates from the skin, only 63% showed susceptibility to clindamycin, a drug of choice for treating drug-resistant skin infections (pyoderma) in dogs, and 95% *Staphylococcus* spp. isolates from the ear were susceptible to amoxicillin–clavulanic acid. In comparison to our findings, a study conducted in Japan found 82% of all the *Staphylococcus pseudintermedius* isolates from dogs with otitis externa were susceptible to amoxicillin–clavulanic acid [18]. In our antibiograms, no *Staphylococcus* spp. isolate from the ear was susceptible to penicillin. A study conducted in France reported a 68.5% resistance to penicillin among *Staphylococcus pseudintermedius* isolated from dogs with otitis [19]. Sixty-nine percent of the 71 *E. coli* isolated from urine were susceptible to ticarcillin and 73% of the 45 *Enterococcus* spp. isolated from urine were susceptible to amoxicillin–clavulanic acid and ampicillin. Our urinary antibiogram showed lower susceptibility of *Enterobacter* spp. to amoxicillin–clavulanic acid when compared to the findings of an Australian study [20] that reported a 95% susceptibility of *Enterobacter faecalis* to amoxicillin–clavulanic acid. Compared to ours, the Australian study had a larger sample size collected over a five-year period. We observed from our antibiograms that a 100% of the canine *E. coli* isolated from a veterinary teaching hospital in Indiana in 2015 were susceptible to imipenem. This is similar to a previous report from Marshfield labs, where a 100% susceptibility of canine *E. coli* isolates tested in the years 2017 through 2018 were susceptible to imipenem [21]. Additionally, a previous study reported a 99.9% cumulative susceptibility of *E. coli* isolates from human samples to imipenem in Pennsylvania [15]. Resistance to carbapenems (such as imipenem) among veterinary isolates is rare, and imipenem is highly stable against most β-lactamases [22]. Possibly, the 100% susceptibility observed in our antibiograms is due to the pharmacodynamic properties of imipenem or could be because imipenem is not frequently used and is not a first-line drug in the treatment of dogs with bacterial infections. However, it is important to note that the reported proportions of susceptible isolates could be subject to selection bias because selection bias can be an issue in clinical samples submitted to a diagnostic laboratory.

Having an insufficient number of isolates for analysis is a concern when creating antibiograms in smaller human hospitals and for infrequently isolated bacteria [11]. In our analysis, the number of isolates for other animal species (e.g., cattle) and for some individual bacterial species was very low (fewer than 30 isolates), and therefore antibiograms were not constructed. Depending upon the patient population and the frequency of clinical sample submission, we observe that construction of veterinary antibiograms for the various veterinary animal species (except for dogs) and for some bacterial species will be challenging due to the relatively small number of samples submitted for bacterial culture and antimicrobial susceptibility testing. This challenge could be overcome with the use of data collected over a longer period of time.

Periodic preparation and distribution of cumulative antibiograms to clinicians, the hospital pharmacist, and the infection-control personnel at this hospital in the form of pocket guides and website postings will be useful. The impact of the antibiograms on veterinary prescription practices through periodic assessment of hospital pharmacy records and via surveys periodically administered to clinicians should be routinely assessed.

## 4. Materials and Methods

We used laboratory data for the period from 1 January 2015 to 31 December 2015 provided by the Animal Disease Diagnostic Lab (ADDL) at a veterinary teaching hospital in Indiana, the United States. The data provided did not contain personally identifiable information about the clinician or client and was provided in a Microsoft Excel file. These data consisted of 661 (complete and partial) antimicrobial susceptibility profiles for isolates obtained from dogs (*n* = 578), cats (*n* = 68), rats (*n* = 4), wolves (*n* = 2), guinea pigs (*n* = 2), chinchillas (*n* = 2), fox (*n* = 1), conure (*n* = 1), and snake (*n* = 1). No animal species was recorded for two isolates. The data were verified and examined for completeness. The data captured in the dataset included the hospital section where the patient was admitted, the location from which the sample was collected, the bacterial isolate identified, and the antimicrobial susceptibility profiles for various antimicrobials. Given the low numbers reported for the various animal species, only the dog data were utilized in the construction of the antibiograms. In total, 563 canine antimicrobial susceptibility profiles were analyzed.

We used the Clinical and Laboratory Standards Institute (CLSI) guidelines for the use in the construction of the antibiograms in human healthcare settings as the basis for the veterinary antibiograms [23]. In brief, the CLSI recommends that the isolates included in antibiogram construction are identified during diagnostic sampling and include only the results from the first isolate recovered from each patient. Information from individual bacterial species should only be presented if there are at least 30 isolates tested. In addition, data should be presented as the total percent of the isolates susceptible to drugs that are routinely tested and should not include the percentage of isolates with intermediate susceptibility results. Data included in hospital antibiograms should be updated annually [11].

In our analysis, we utilized only the first isolate of a given bacterial species recovered from each canine patient during 2015 in the calculation of percent susceptibility for each organism/antimicrobial combination. Antibiograms for drugs that are routinely tested were prepared and presented in tabular form. These drugs include amikacin, amoxicillin/clavulanic acid, ampicillin, cefazolin, cefovecin, cefoxitin, cefpodoxime, ceftiofur, cephalothin, chloramphenicol, clindamycin, doxycycline, enrofloxacin, erythromycin, gentamycin, imipenem, marbofloxacin, oxacillin + 2%NACL, penicillin, ticarcillin, ticarcillin/clavulanic acid, trimethoprim/sulfamethoxazole.

Only the percentage of susceptible isolates and not those which are of intermediate susceptibility were presented. Organisms with fewer than 30 isolates were only included if their inclusion was deemed essential based on existing knowledge showing that resistance against common antimicrobials is possible [22]. In some instances, species within a genus with fewer than 30 individual isolates were grouped together. In cases where the number of isolates used to calculate the percent susceptibility to a given drug were fewer compared to other drugs, a note was appended to indicate that the calculation of percent susceptibility was conducted using fewer isolates. The completed antibiograms were further validated to ensure that only percent susceptibilities for antimicrobial agents that are appropriate for clinical use in the species were reported. However, percent susceptibilities for known intrinsically resistant pathogens such as *Pseudomonas aeruginosa* and *Enterococcus* spp. were reported in the antibiograms to guide the antimicrobial selection. Additionally, percentage susceptibilities for drugs primarily used to treat Gram-positive infections (lincosamides, e.g., clindamycin and macrolides, e.g., erythromycin) were reported for Gram-negative pathogens with a footnote added stating that these two drug classes are primarily used against Gram-positive organisms. The percentage susceptibilities were calculated using commercial statistical software (SAS, version 9.4, SAS Institute Inc, Cary, NC, USA) and the steps taken in constructing the antibiograms are given in Figure 1.

## 5. Conclusions

Creating cumulative antibiograms for dogs seen at a veterinary teaching hospital is feasible and practical. Susceptibility differences to specific antimicrobials were identified among bacteria isolated at a veterinary teaching hospital in Indiana from January 2015 through December 2015. Cumulative antibiograms could be useful in guiding empiric antimicrobial therapy and for detecting and monitoring the AMR trends at this teaching hospital.

## Figures and Tables

**Figure 1 antibiotics-12-01034-f001:**
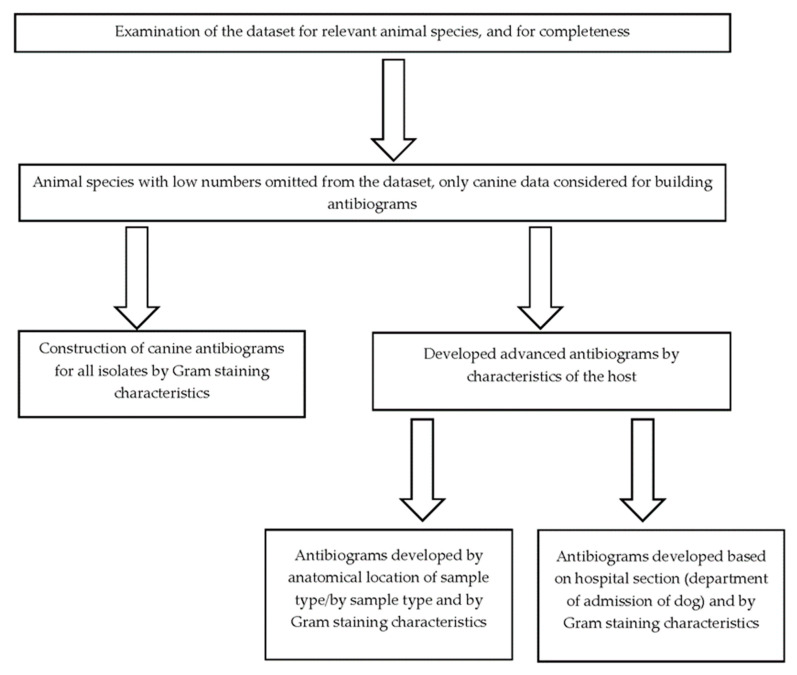
The algorithm used to construct veterinary antibiograms for dogs presented to a veterinary teaching hospital in Indiana, the United States, 1 January 2015 to 31 December 2015.

**Table 1 antibiotics-12-01034-t001:** A general antibiogram showing percent antimicrobial susceptibility of all Gram-negative bacteria isolated from a veterinary teaching hospital in Indiana, the United States in 2015 and grouped at genus/species level.

	*Escherichia coli*	*Pseudomonas* spp.	*Proteus mirabilis*
Number of Isolates	100	27	26
Antimicrobial			
Amikacin	99%	100%	92%
Amoxicillin/clavulanic acid	78%	4%	100%
Ampicillin	64%	4%	84%
Cefazolin	79%	4%	88%
Cefovecin	82%	7%	96%
Cefoxitin	83%	4%	96%
Cefpodoxime	82%	4%	96%
Ceftiofur	79%	7%	96%
Cephalothin **	100%		100%
Chloramphenicol	84%	4%	69%
Clindamycin ^¥^	2%	4%	0%
Doxycycline	78%	7%	0%
Enrofloxacin	85%	70%	92%
Erythromycin ^¥^	10%	4%	0%
Gentamycin	94%	85%	85%
Imipenem	100%	100%	96%
Marbofloxacin	83%		92%
Oxacillin + 2%NACL	0%	7%	0%
Penicillin	0%	4%	0%
Ticarcillin	64%	100%	85%
Ticarcillin/clavulanic acid	75%	100%	96%
Trimethoprim/sulfamethoxazole	85%	11%	85%

** Number of isolates tested for cephalothin susceptibility were as follows: *E. coli* = 67, *Proteus mirabilis* = 20. ^¥^ indicates the drug is primarily used for Gram-positive organisms. Grey-shading indicates that susceptibility to a drug was not tested with that organism.

**Table 2 antibiotics-12-01034-t002:** A general antibiogram showing percent antimicrobial susceptibility of Gram-positive bacteria isolated from a veterinary teaching hospital in Indiana, the United States in 2015 and grouped at genus level.

	*Staphylococcus* spp.	*Streptococcus* spp.	*Corynebacterium* spp.	*Enterococcus* spp.
Number of Isolates	170	64	30	65
Antimicrobial				
Amikacin	89%	84%	93%	0%
Amoxicillin/clavulanic acid	84%	100%	59%	72%
Ampicillin	16%	95%	38%	72%
Cefazolin	85%	100%	50%	0%
Cefovecin	80%	95%	48%	0%
Cefoxitin	84%	95%	41%	0%
Cefpodoxime	80%	95%	38%	0%
Ceftiofur	81%	97%	53%	0%
Cephalothin **	100%	100%		
Chloramphenicol	86%	92%	80%	88%
Clindamycin ^¥^	68%	90%	69%	0%
Doxycycline	67%	78%	90%	51%
Enrofloxacin	72%	34%	38%	6%
Erythromycin ^¥^	68%	3%	63%	31%
Gentamycin	76%	92%	87%	2%
Imipenem	85%	100%	59%	74%
Marbofloxacin	79%	60%		
Oxacillin + 2%NACL	85%	98%	21%	6%
Penicillin	0%	92%	0	69%
Ticarcillin	75%	97%	60%	15%
Ticarcillin/clavulanic acid	84%	97%	52%	18%
Trimethoprim/sulfamethoxazole	75%	94%	87%	3%

** Number of isolates tested for cephalothin susceptibility were as follows: *Staphylococcus* spp. = 105, *Streptococcus* spp. = 55. ^¥^ indicates the drug is primarily used for Gram-positive organisms. Grey-shading indicates that susceptibility to a drug was not tested with that organism.

**Table 3 antibiotics-12-01034-t003:** Antibiogram showing percent antimicrobial susceptibility of Gram-positive bacteria isolated from the dermatology, surgery and oncology sections of a veterinary teaching hospital in Indiana, the United States, 2015.

Dermatology			Surgery		Oncology			
	*Staphylococcus* spp.	*Streptococcus* spp.		*Staphylococcus* spp.		*Staphylococcus* spp.	*Streptococcus* spp.	*Enterococcus* spp.
Number of Isolates	41	12	Number of Isolates	20	Number of Isolates	33	20	17
Antimicrobial								
Amikacin	93%	75%	Amikacin	85%	Amikacin	91%	85%	0%
Amoxicillin/clavulanic acid	85%	100%	Amoxicillin/clavulanic acid	65%	Amoxicillin/clavulanic acid	85%	100%	65%
Ampicillin	^ⱡ^ 10%	100%	Ampicillin *	13%	Ampicillin	24%	95%	65%
Cefazolin	85%	100%	Cefazolin	65%	Cefazolin	85%	100%	0%
Cefovecin	78%	100%	Cefovecin	60%	Cefovecin	85%	100%	0%
Cefoxitin	85%	100%	Cefoxitin	65%	Cefoxitin	85%	95%	0%
Cefpodoxime	78%	100%	Cefpodoxime	60%	Cefpodoxime	85%	100%	0%
Ceftiofur	85%	100%	Ceftiofur	60%	Ceftiofur	79%	100%	0%
Cephalothin **	100%	100%	Cephalothin **	100%	Cephalothin **	100%	100%	
Chloramphenicol	88%	100%	Chloramphenicol	85%	Chloramphenicol	76%	95%	88%
Clindamycin	68%	92%	Clindamycin	55%	Clindamycin	67%	95%	0%
Doxycycline	71%	92%	Doxycycline	90%	Doxycycline	61%	75%	47%
Enrofloxacin	68%	42%	Enrofloxacin	60%	Enrofloxacin	76%	25%	0%
Erythromycin	71%	0%	Erythromycin	55%	Erythromycin	67%	5%	35%
Gentamycin	66%	92%	Gentamycin	70%	Gentamycin	82%	95%	0%
Imipenem	85%	100%	Imipenem	70%	Imipenem	85%	100%	71%
Marbofloxacin	71%	67%	Marbofloxacin	75%	Marbofloxacin	85%	50%	
Oxacillin + 2%NACL	85%	100%	Oxacillin + 2%NACL	70%	Oxacillin + 2%NACL	85%	100%	6%
Penicillin *	0%	100%	Penicillin *	0%	Penicillin *	0%	95%	65%
Ticarcillin *	79%	100%	Ticarcillin *	44%	Ticarcillin *	80%	100%	24%
Ticarcillin/clavulanic acid	85%	100%	Ticarcillin/clavulanic acid	65%	Ticarcillin/clavulanic acid	85%	100%	24%
Trimethoprim/sulfamethoxazole	68%	100%	Trimethoprim/sulfamethoxazole	80%	Trimethoprim/sulfamethoxazole	79%	95%	0%

Dermatology section: ^ⱡ^ Number of isolates tested: *Staphylococcus* spp. = 29. ** Number of isolates tested for cephalothin susceptibility were as follows: *Staphylococcus* spp. = 26, *Streptococcus* spp. = 11. * Number of *Staphylococcus* spp. isolates tested for penicillin and ticarcillin susceptibility was 29. Surgery section: * Number of *Staphylococcus* spp. isolates tested for ampicillin; penicillin and ticarcillin susceptibility was 16. ** Number of isolates tested for cephalothin susceptibility was 10. Oncology section: ** Numbers of isolates tested for cephalothin susceptibility were as follows: *Staphylococcus* spp. = 20, *Streptococcus* spp. = 19. * Number of *Staphylococcus* spp. isolates for penicillin and ticarcillin susceptibility was 25. Grey-shading indicates that susceptibility to a drug was not tested with that organism.

**Table 4 antibiotics-12-01034-t004:** Antibiogram showing percent antimicrobial susceptibility of Gram-positive bacteria isolated from the medicine and emergency sections of a veterinary teaching hospital in Indiana, the United States, 2015.

Medicine			Emergency		
	*Staphylococcus* spp.	*Enterococcus* spp.		*Staphylococcus* spp.	*Enterococcus* spp.
Number of Isolates	28	15	Number of Isolates	16	13
Antimicrobial					
Amikacin	86%	0%	Amikacin	88%	0%
Amoxicillin/clavulanic acid	82%	80%	Amoxicillin/clavulanic acid	94%	69%
Ampicillin	* 6%	80%	Ampicillin *	13%	69%
Cefazolin	82%	0%	Cefazolin	94%	0%
Cefovecin	79%	0%	Cefovecin	88%	0%
Cefoxitin	82%	0%	Cefoxitin	94%	0%
Cefpodoxime	79%	0%	Cefpodoxime	88%	0%
Ceftiofur	79%	0%	Ceftiofur	88%	0%
Cephalothin **	100%		Cephalothin **	100%	
Chloramphenicol	86%	100%	Chloramphenicol	94%	62%
Clindamycin	61%	0%	Clindamycin	69%	0%
Doxycycline	54%	60%	Doxycycline	75%	31%
Enrofloxacin	71%	7%	Enrofloxacin	69%	8%
Erythromycin	61%	27%	Erythromycin	69%	8%
Gentamycin	75%	7%	Gentamycin	81%	0%
Imipenem	82%	80%	Imipenem	94%	695
Marbofloxacin	71%		Marbofloxacin	88%	
Oxacillin + 2%NACL	82%	7%	Oxacillin + 2%NACL	94%	0%
Penicillin	* 0%	73%	Penicillin	* 0%	69%
Ticarcillin	* 72%	13%	Ticarcillin	* 88%	0%
Ticarcillin/clavulanic acid	82%	20%	Ticarcillin/clavulanic acid	94%	0%
Trimethoprim/sulfamethoxazole	61%	7%	Trimethoprim/sulfamethoxazole	81%	0%

Medicine section: * Number of *Staphylococcus* spp. isolates for ampicillin, penicillin and ticarcillin susceptibility was 18. ** Number of *Staphylococcus* spp. isolates for cephalothin susceptibility was 16. Emergency section: * Number of *Staphylococcus* spp. isolates for ampicillin, penicillin and ticarcillin susceptibility was 8. ** Number of *Staphylococcus* spp. isolates for cephalothin susceptibility was 13. Grey-shading indicates that susceptibility to a drug was not tested with that organism.

**Table 5 antibiotics-12-01034-t005:** Antibiogram showing percent antimicrobial susceptibility of Gram-negative bacteria isolated from different sections of a veterinary teaching hospital in Indiana, the United States, 2015.

Oncology		Medicine		Emergency	
*Escherichia coli*		*Escherichia coli*		*Escherichia coli*	
Number of Isolates	28	Number of Isolates	26	Number of Isolates	15
Antimicrobial					
Amikacin	100%	Amikacin	100%	Amikacin	100%
Amoxicillin/clavulanic acid	71%	Amoxicillin/clavulanic acid	81%	Amoxicillin/clavulanic acid	67%
Ampicillin	57%	Ampicillin	69%	Ampicillin	73%
Cefazolin	79%	Cefazolin	85%	Cefazolin	73%
Cefovecin	82%	Cefovecin	88%	Cefovecin	73%
Cefoxitin	86%	Cefoxitin	85%	Cefoxitin	73%
Cefpodoxime	82%	Cefpodoxime	88%	Cefpodoxime	73%
Ceftiofur	82%	Ceftiofur	81%	Ceftiofur	67%
Cephalothin **	100%	Cephalothin **	100%	Cephalothin **	9%
Chloramphenicol	86%	Chloramphenicol	85%	Chloramphenicol	93%
Clindamycin ^¥^	0%	Clindamycin	4%	Clindamycin	7%
Doxycycline	75%	Doxycycline	81%	Doxycycline	87%
Enrofloxacin	82%	Enrofloxacin	85%	Enrofloxacin	93%
Erythromycin ^¥^	11%	Erythromycin	12%	Erythromycin	7%
Gentamycin	93%	Gentamycin	96%	Gentamycin	100%
Imipenem	100%	Imipenem	100%	Imipenem	100%
Marbofloxacin	82%	Marbofloxacin	85%	Marbofloxacin	93%
Oxacillin + 2%NACL	0%	Oxacillin + 2%NACL	0%	Oxacillin + 2%NACL	0%
Penicillin	0%	Penicillin	0%	Penicillin	0%
Ticarcillin	57%	Ticarcillin	69%	Ticarcillin	73%
Ticarcillin/clavulanic acid	71%	Ticarcillin/clavulanic acid	80%	Ticarcillin/clavulanic acid	73%
Trimethoprim/sulfamethoxazole	79%	Trimethoprim/sulfamethoxazole	88%	Trimethoprim/sulfamethoxazole	100%

Oncology section: ** Number of isolates for cephalothin susceptibility was 19. Medicine section: ** Number of isolates for cephalothin susceptibility was 18. Emergency section: ** Number of isolates for cephalothin susceptibility was 9. ^¥^ indicates the drug is primarily used for Gram-positive organisms. Grey-shading indicates that susceptibility to a drug was not tested with that organism.

**Table 6 antibiotics-12-01034-t006:** Antibiogram showing percent antimicrobial susceptibility of *Staphylococcus* spp. isolated from samples obtained from the ear and skin of canine patients at a veterinary teaching hospital in Indiana, the United States, 2015.

Samples from the Ear		Samples from the Skin	
*Staphylococcus* spp.		*Staphylococcus* spp.	
Number of Isolates	21	Number of Isolates	43
Antimicrobial			
Amikacin	90%	Amikacin	93%
Amoxicillin/clavulanic acid	95%	Amoxicillin/clavulanic acid	88%
Ampicillin *	29%	Ampicillin *	3%
Cefazolin	95%	Cefazolin	88%
Cefovecin	90%	Cefovecin	81%
Cefoxitin	95%	Cefoxitin	88%
Cefpodoxime	90%	Cefpodoxime	81%
Ceftiofur	95%	Ceftiofur	88%
Cephalothin **	100%	Cephalothin **	100%
Chloramphenicol	95%	Chloramphenicol	84%
Clindamycin	81%	Clindamycin	63%
Doxycycline	71%	Doxycycline	65%
Enrofloxacin	76%	Enrofloxacin	70%
Erythromycin	81%	Erythromycin	65%
Gentamycin	86%	Gentamycin	67%
Imipenem	95%	Imipenem	88%
Marbofloxacin	81%	Marbofloxacin	72%
Oxacillin + 2%NACL	95%	Oxacillin + 2%NACL	88%
Penicillin *	0%	Penicillin *	0%
Ticarcillin *	93%	Ticarcillin *	81%
Ticarcillin/clavulanic acid	95%	Ticarcillin/clavulanic acid	88%
Trimethoprim/sulfamethoxazole	71%	Trimethoprim/sulfamethoxazole	63%

Ear samples: * Number of *Staphylococcus* spp. isolates for ampicillin, penicillin and ticarcillin susceptibility was 14. ** Number of *Staphylococcus* spp. isolates for cephalothin susceptibility was 17. Skin samples: * Number of *Staphylococcus* spp. isolates for ampicillin, penicillin and ticarcillin susceptibility was 31. ** Number of *Staphylococcus* spp. isolates for cephalothin susceptibility was 26. Grey-shading indicates that susceptibility to a drug was not tested with that organism.

**Table 7 antibiotics-12-01034-t007:** Antibiogram showing percent antimicrobial susceptibility of bacteria isolated from urine samples of canine patients at a veterinary teaching hospital in Indiana, the United States, 2015.

Gram-Negative				Gram-Positive			
	*Escherichia coli*	*Pseudomonas* spp.	*Proteus mirabilis*		*Staphylococcus* spp.	*Streptococcus* spp.	*Enterococcus* spp.
Number of Isolates	71	13	14	Number of Isolates	49	27	45
Antimicrobial							
Amikacin	99%	100%	93%	Amikacin	90%	89%	0%
Amoxicillin/clavulanic acid	80%	8%	100%	Amoxicillin/clavulanic acid	86%	100%	73%
Ampicillin	69%	8%	86%	Ampicillin	23%	93%	73%
Cefazolin	82%	8%	93%	Cefazolin	86%	100%	0%
Cefovecin	86%	15%	93%	Cefovecin	82%	96%	0%
Cefoxitin	84%	8%	93%	Cefoxitin	86%	93%	0%
Cefpodoxime	84%	8%	93%	Cefpodoxime	82%	93%	0%
Ceftiofur	80%	15%	93%	Ceftiofur	76%	96%	0%
Cephalothin **	100%		100%	Cephalothin **	100%	100%	
Chloramphenicol	83%	8%	71%	Chloramphenicol	78%	89%	89%
Clindamycin ^¥^	1%	8%	0%	Clindamycin	69%	96%	0%
Doxycycline	82%	15%	0%	Doxycycline	59%	78%	49%
Enrofloxacin	87%	77%	93%	Enrofloxacin	76%	26%	4%
Erythromycin ^¥^	11%	8%	0%	Erythromycin	69%	7%	31%
Gentamycin	94%	100%	86%	Gentamycin	80%	92%	2%
Imipenem	100%	100%	100%	Imipenem	86%	100%	76%
Marbofloxacin	86%		93%	Marbofloxacin	82%	48%	
Oxacillin + 2%NACL	0%	15%	0%	Oxacillin + 2%NACL	86%	100%	7%
Penicillin	0%	8%	0%	Penicillin *	0%	89%	69%
Ticarcillin	69%	100%	86%	Ticarcillin *	80%	96%	13%
Ticarcillin/clavulanic acid	79%	100%	93%	Ticarcillin/clavulanic acid	86%	96%	16%
Trimethoprim/sulfamethoxazole	86%	23%	93%	Trimethoprim/sulfamethoxazole	80%	89%	4%

Gram-negatives: ** Numbers of isolates tested for cephalothin susceptibility were as follows: *E. coli* = 50, *Proteus mirabilis* = 12. ^¥^ indicates the drug is primarily used for Gram-positive organisms. Gram-positives: ** Numbers of isolates tested for cephalothin susceptibility were as follows: *Staphylococcus* spp. = 29, *Streptococcus* spp. = 24. * Number of *Staphylococcus* spp. isolates for penicillin and ticarcillin susceptibility was 35. Grey-shading indicates that susceptibility to a drug was not tested with that organism.

## Data Availability

The data for this study are available from the corresponding author upon reasonable request.

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
