# Peer review of "Approaches Used to Construct Antibiograms for Dogs in a Veterinary Teaching Hospital in the United States"

_antibiotics, 2023, doi:10.3390/antibiotics12061034_

Round 1
Reviewer 1 Report
In the manuscript "Approaches used to construct antibiograms for dogs in a veterinary teaching hospital in the United States" the authors examine the practicality of constructing antibiograms for veterinary practice.
The data presented in this manuscript provide a description of a practical approach in making antibiograms for the veterinary practice. The problem of antimicrobial resistance is global and I strongly believe that studies like this contributes to the better understanding of the approach we should use to deal with it in veterinary clinics.
I strongly recommend that the authors should also use data from other studies outside USA (e.g Australia, Europe or Asia if available) in "Discussion" section for a more complete view regarding antimicrobial resistance problem.
Author Response
In the manuscript "Approaches used to construct antibiograms for dogs in a veterinary teaching hospital in the United States" the authors examine the practicality of constructing antibiograms for veterinary practice.
The data presented in this manuscript provide a description of a practical approach in making antibiograms for the veterinary practice. The problem of antimicrobial resistance is global and I strongly believe that studies like this contributes to the better understanding of the approach we should use to deal with it in veterinary clinics.
Response: We thank you so much for taking your time to review our manuscript and for providing feedback for improving our manuscript.
I strongly recommend that the authors should also use data from other studies outside USA (e.g Australia, Europe or Asia if available) in "Discussion" section for a more complete view regarding antimicrobial resistance problem.
Response: We thank you for this suggestion. We have added a discussion of studies conducted in Europe, Japan, and Australia. Please see lines 199-213 in the discussion section of the revised manuscript.
Reviewer 2 Report
The work by John E. Ekakoro and colleagues show in this study, the practicality of constructing cumulative antibiograms for a veterinary tertiary care hospital. This paper describes and documents the approaches used, and the challenges that were encountered in constructing cumulative antibiograms for a veterinary teaching hospital in Indiana, United States.
I have found some minor details in the ms that need to be revised, which hopefully will improve the quality of the present work.
. 1) In the description of the figures, abbreviations should not be used. 2) Keywords: Please do not include words mentioned in the title of the ms.
3) Why did you use 563 dogs in this study
Author Response
The work by John E. Ekakoro and colleagues show in this study, the practicality of constructing cumulative antibiograms for a veterinary tertiary care hospital. This paper describes and documents the approaches used, and the challenges that were encountered in constructing cumulative antibiograms for a veterinary teaching hospital in Indiana, United States.
I have found some minor details in the ms that need to be revised, which hopefully will improve the quality of the present work.
Response: We thank you so much for reviewing our manuscript and providing feedback that has improved our manuscript. Thank you again.
- In the description of the figures, abbreviations should not be used.
Response: We have revised the manuscript and removed the abbreviations in the tables. The figure has no abbreviations as well.
- Keywords: Please do not include words mentioned in the title of the ms.
Response: We have revised the key words as suggested. Please see lines 32-33 in the revised manuscript.
- Why did you use 563 dogs in this study
Response: This was the total number of isolates submitted from dogs that had complete records during the study period. We mention in line 248 the total number of records frm dogs, in line 250 of the revised manuscript we mention that we examined the data for completeness and we have added in line 255 that we used a total of 563 samples.
Reviewer 3 Report
Comment to “Approaches used to construct antibiograms for dogs in a veterinary teaching hospital in the United States”
The article aims to describe the approaches that were used to construct antibiograms from clinical samples collected from dogs seen at a veterinary teaching hospital. The methods described can be useful in guiding veterinary antibiogram development for empiric therapy.
The article is useful and important. The suggested questions are listed below.
1. Tables should add the abbreviation as the notes;
2. Though material and methods were listed, they are simple and details were not presented.
Author Response
Comment to “Approaches used to construct antibiograms for dogs in a veterinary teaching hospital in the United States”
The article aims to describe the approaches that were used to construct antibiograms from clinical samples collected from dogs seen at a veterinary teaching hospital. The methods described can be useful in guiding veterinary antibiogram development for empiric therapy.
Response: We thank you so much for reviewing our manuscript and providing feedback that has improved our manuscript. Thank you again.
The article is useful and important. The suggested questions are listed below.
- Tables should add the abbreviation as the notes;
Response: We thank you for this observation. We have addressed this by revising the manuscript accordingly as suggested.
- Though material and methods were listed, they are simple, and details were not presented.
Response: We agree that we have described the materials and methods that we used in a precise and concise manner. All details that would be necessary in guiding the development of veterinary antibiograms have been included.